# Personalized Immunotherapies for Type 1 Diabetes: Who, What, When, and How?

**DOI:** 10.3390/jpm12040542

**Published:** 2022-03-29

**Authors:** Claire Deligne, Sylvaine You, Roberto Mallone

**Affiliations:** 1Institut Cochin, CNRS, INSERM, Université de Paris, 75014 Paris, France; sylvaine.you@inserm.fr (S.Y.); roberto.mallone@inserm.fr (R.M.); 2Assistance Publique Hôpitaux de Paris, Service de Diabétologie et Immunologie Clinique, Cochin Hospital, 75014 Paris, France

**Keywords:** type 1 diabetes, autoimmunity, autoantigens, endotypes, biomarkers, immune tolerance, immunotherapy, personalized medicine

## Abstract

Our understanding of the immunopathological features of type 1 diabetes (T1D) has greatly improved over the past two decades and has shed light on disease heterogeneity dictated by multiple immune, metabolic, and clinical parameters. This may explain the limited effects of immunotherapies tested so far to durably revert or prevent T1D, for which life-long insulin replacement remains the only therapeutic option. In the era of omics and precision medicine, offering personalized treatment could contribute to turning this tide. Here, we discuss how to structure the selection of the right patient at the right time for the right treatment. This individualized therapeutic approach involves enrolling patients at a defined disease stage depending on the target and mode of action of the selected drug, and better stratifying patients based on their T1D endotype, reflecting intrinsic disease aggressiveness and immune context. To this end, biomarker screening will be critical, not only to help stratify patients and disease stage, but also to select the best predicted responders ahead of treatment and at early time points during clinical trials. This strategy could contribute to increase therapeutic efficacy, notably through the selection of drugs with complementary effects, and to further develop precision multi-hit medicine.

## 1. Introduction

Type 1 diabetes (T1D) is an autoimmune disease during which a tolerance break leads to the destruction of pancreatic β cells by cellular immunity, leading to a life-long dependency on exogenous insulin. Our understanding of the triggers involved in T1D pathogenesis has greatly evolved [1,2] since its recognition as an autoimmune disease in 1974 [3]. It results from a complex interplay between genetic predispositions, such as those encoded by human leukocyte antigen (HLA) class I and II, insulin, and PTPN22 polymorphisms [4,5] and candidate environmental cues including enterovirus (e.g., Coxsackievirus) exposure, gut dysbiosis, and overweight, among others [6]. T1D prediction and diagnosis have been established for decades and are based on the detection of autoantibodies (aAbs) directed against β-cell proteins such as insulin (so-called insulin aAbs, IAA), glutamic acid decarboxylase (GAD), insulinoma-associated protein 2 (IA-2), and/or zinc transporter 8 (ZnT8) [7], which precede clinical symptoms by years. Together with HLA class II genotyping and measurements of insulin secretion and blood glucose, they constitute the triad of T1D staging.

To date, there is no cure for T1D, and the chronic administration of insulin is only a replacement therapy still associated with a decreased life expectancy due to microvascular and macrovascular complications [8,9,10]. Despite intensive efforts during the past 30 years to develop disease-modifying therapies, no clinical trial has so far succeeded in inducing T1D remission after clinical onset or preventing clinical progression in at-risk patients. A particular emphasis has been put on immunotherapies through different strategies, including B-cell depletion, T-cell modulation, cytokine targeting, and antigen-specific tolerance induction. Overall, these strategies still fall short in achieving durable therapeutic benefits. These failures are open to several explanations, such as a late patient enrollment into a majority of these trials, at a time when most β cells are already destroyed, and a disease heterogeneity that has been largely overlooked. There are indeed several T1D “endotypes”, exhibiting distinct clinical and biological features (age of onset, genetic polymorphisms, severity and composition of the immune infiltrates, magnitude of β-cell destruction) [11].

In this review, we will discuss how the heterogeneity of T1D pathophysiology may dictate the choice and response to immunotherapies, and how tailoring the right treatment to the right patient at the right time can turn the tide toward clinical benefit by halting the autoimmune β-cell destruction. Moving to more personalized medicine in T1D implies a better categorization and monitoring of disease progression and aggressiveness identified by sets of biomarkers that will provide guidance to “when, who, how, and what” before launching T1D intervention or prevention trials.

## 2. Disease Stage (“When”)

Most patients recruited in T1D clinical trials exhibit a symptomatic hyperglycemia associated with an advanced destruction of their pancreatic β-cell mass—so-called stage 3 (clinical) disease. Thus, by the time of intervention, the damage is already done. In contrast, stage 1 patients, who are normoglycemic but positive for at least 2 aAbs, and stage 2 patients, who present dysglycemia (e.g., transient hyperglycemic peaks after a glucose challenge), exhibit humoral and cellular islet autoimmunity, but their β-cell mass and function are still preserved, which provides the opportunity to reorient the immune responses towards tolerance before irreversible damage. It is henceforth accepted that adapting and searching for the best therapeutic window for a given treatment represents a first step towards personalized intervention [12]. The best example is given by non Fc-binding humanized anti-CD3 monoclonal antibodies (mAbs; teplizumab and otelixizumab) inducing a partial depletion of effector T cells and an anergic/exhausted phenotype. These features were initially brought to light in clinical trials showing their capacity to preserve the residual insulin response (measured as C-peptide) in new-onset T1D patients [13,14]. The success of these studies appears, however, modest in comparison to the therapeutic efficacy obtained when teplizumab was administered over a 2-week course to at-risk aAb^+^ patients with stage 2 disease, leading to an impressive 2-year delay in T1D onset (48.4 versus 24.4 months in the placebo group) [15]. Defining the best window of intervention is of importance given the number of therapies that aimed at reversing the course of T1D in recently diagnosed patients, and only showed partial and transient preservation of endogenous β-cell function, including abatacept (immunomodulatory CTLA4-Ig fusion protein) [16], alefacept (LFA3-Ig fusion protein) [17], anti-thymocyte globulin (ATG) [18], rituximab (anti-CD20 mAb) [19], and golimumab (anti-tumor necrosis factor(TNF)-α) [20]. Results obtained with teplizumab encourage applications of these immunomodulatory strategies at the pre-diabetic stage, and several studies are currently ongoing in at-risk patients with rituximab (NCT03929601) or abatacept (NCT01773707) [21]. Thus, we may speculate that the sooner the intervention, the greater the benefit for the patients in terms of disease prevention and progression, through enhanced induction of immune tolerance mechanisms and reduced damage to the pancreatic islets. It may, however, not be so straightforward, as a greater protection from clinical progression to T1D was observed within the first 3 years after administration of teplizumab administration in aAb^+^ patients [15]. In other words, teplizumab had a greater effect in rapid progressors than in slower ones, and the response was also superior in patients exhibiting the lowest C-peptide levels at enrollment. These results suggest that teplizumab efficacy is optimal in individuals with high risk of progression, rather than in subjects selected at an earlier disease stage. These observations are consistent with observations in the non-obese diabetic (NOD) mouse model of T1D, showing the therapeutic potential of anti-CD3 Ab applied at diabetes onset or at the late pre-diabetic stage, but not when the treatment is initiated earlier [22].

The crucial importance of intervening at the right time has been brought a step further with the pre-POINT trial, which ultimately aimed at assessing the efficacy of orally administrated insulin to prevent or delay T1D onset in aAb-negative, genetically at-risk children [23]. The rationale for targeting this pediatric population before any sign of autoimmunity is that, so far, all clinical trials using tolerogenic vaccination with β-cell-derived antigens have failed, both in new-onset diabetic and in at-risk aAb^+^ patients [24]. This is probably due to a late timing of treatment, i.e., after appearance of multiple aAbs and initiation of β-cell destruction, marking an advanced and active autoimmunity stage that cannot be properly controlled. The pre-POINT phase 1/2 trial confirmed the safety of oral insulin administration in these “stage 0” patients aged 2–7 years (no severe hypoglycemic events, no disease precipitation), and showed the induction of insulin-reactive CD4^+^ T cells compatible with an immunoregulatory skewing. In line with the notion that islet aAbs frequently appear during the first 2 years of life [25,26], a pre-POINT Early study (NCT02547519) was launched in 6-month-old to 3-year-old aAb-negative at-risk children [27]. High dose oral insulin was safe but failed to demonstrate immune benefits as compared to the placebo group. The ongoing phase 2/3 POINT trial recruiting 4–7-month-old children should allow for the consolidation of safety and assessment of efficacy (NCT03364868).

In this context, we also tested this “the sooner the better” concept by treating in utero fetuses in a diabetic-prone mouse model. Systemic administration to pregnant mice of preproinsulin tagged with an IgG1 Fc domain (PPI-Fc) led to a 3-fold decrease in diabetes incidence in the offspring by neonatal Fc receptor (FcRn)-mediated materno–fetal transmission of the antigen [28]. We confirmed this result by demonstrating that oral administration of PPI-Fc directly to newborn mice delayed disease onset in the NOD. *Insulin2*^−/−^ aggressive model of spontaneous diabetes [29]. These results support the idea that early T1D prevention could be achieved in selected patients. This also lends credibility to the increasing screening efforts in the general pediatric population for the presence of islet aAbs, as undertaken during well-baby visits in over 90,000 children aged 2 to 5 years between 2015 and 2019 in the German Fr1da study [30]. Together with an increasing availability of aAb screening tests in children [31], large-scale screening could not only help with identifying children at risk of developing T1D at a very early stage, but also provide the infrastructure for prevention trials.

Choosing a therapeutic protocol according to the targeted disease stage is another key point, as the cellular and molecular cues driving the autoimmune response are far from being uniform along T1D progression. If it is accepted that CD8^+^ T cells are the main culprits leading to β-cell destruction, antigen-presenting cells (APCs) also play a crucial role in disease development. Importantly, the expression of type I interferon (IFN)-inducible genes is increased in the blood of children genetically at risk for T1D, before the appearance of circulating islet aAbs, as compared to controls [32,33], but not in patients with an established disease, indicating that innate type I IFN signaling pathways are active very early, at the presymptomatic stage. Given the importance of type I IFN in T1D [34], determination of the time-dependent role of these cytokines is therapeutically crucial, the more so since the Janus kinase (JAK) inhibitor baricitinib, a small molecule targeting the JAK transcription factor downstream of the IFN-α receptor (IFNAR), is currently being investigated in new-onset T1D patients (NCT04774224). Thus, we may hypothesize that targeting type I IFN would be more effective during the very first steps of T1D pathogenesis, when this pathway is most active, opposite to anti-CD3 mAbs that would have a better impact later on, when T cells become engaged in the pancreatic battle. Inaccurate selection of a time window may lead to opposite effects, as exemplified in pre-diabetic 6–8-week-old NOD mice where IFNAR1 blockade with antagonist mAb accelerated diabetes onset [35], while treatment at 2–5 weeks of age (i.e., before any sign of insulitis) prevented it [36]. These differential effects could be explained by the dual, context-dependent role of type I IFN on inflammation, inducing either interleukin (IL)-10 modulatory or pro-inflammatory responses. In very young NOD mice, chronic inflammation in the pancreas is not yet installed, and blocking type I IFN signaling may favor immature or tolerogenic APCs.

Therefore, T1D development and severity are dictated by multiple immune factors (innate responses, B- and T-cell specificity and pathogenicity, cytokine production, β-cell stress and function), whose activation and magnitude define each disease stage. Understanding this intricated network is critical not only to select biomarkers for T1D staging, prognosis, and diagnosis, but also to help select the best immunotherapy (Figure 1).

## 3. Endotypes (“Who”)

The term “endotype” indicates a T1D subtype exhibiting unique functional or biological features [37]. It stems from the important inter-individual differences of the immune and non-immune networks at play in T1D pathophysiology. Altogether, these features can influence trial design, patient recruitment, and therapeutic outcomes, but are rarely used as stratifiers, as diagnosis is solely based on the presence of islet aAbs and low but detectable C-peptide concentrations (to ensure the presence of a residual insulin secretion).

Two T1D endotypes, T1DE1 and T1DE2, have been described based on pancreas histopathology. T1DE1 is characterized by a younger age at diagnosis (<7 years old), few residual insulin-containing islets displaying aberrant proinsulin processing, and abundant immune infiltrates dominated by CD8^+^ T cells and CD20^+^ B cells [38]. Clinically, this T1DE1 histopathological endotype correlates in young children with a rapid decline in C-peptide after diagnosis [39], the early appearance of aAbs first directed toward insulin (so-called “early IAA first” seroconversion), and the HLA-DR4/DQ8 haplotype [40]. In T1DE2, disease develops later in life (≥13 years old) and is associated with significant number of residual insulin-containing islets with preserved proinsulin processing, and less prominent immune infiltrates. Clinically, these older-onset patients are characterized by a slower decline in residual C peptide after diagnosis, by a first seroconversion for anti-GAD aAbs later in life (so-called “late GADA first”) and by the expression of HLA-DR3/DQ2 [41]. These parameters discriminate a more aggressive versus a less aggressive T1D, mirroring the presence of two peaks of incidence according to age at diagnosis, one during childhood and one during adolescence, reinforcing the increasing appreciation that age is the most crucial factor in T1D heterogeneity [1,41]. The presence of islet-infiltrating CD20^+^ B cells is also considered another histopathological biomarker: high B-cell numbers are associated with CD8^+^ T-cell infiltration, lower residual β-cell area, and T1DE1. Indeed, the analysis of T1D pancreatic lesions led to the identification of CD20^hi^ “hyper-immune” versus CD20^low^ “pauci-immune” infiltrates. Patients with clinical onset before 7 years old present a CD20^hi^ phenotype, exhibit severe insulitis [42] with increased numbers of CD8^+^ T cells, and a more extensive loss of β-cell mass than those presenting the CD20^low^ phenotype. RNAseq analyses in new-onset patients similarly identified a prominent B-cell signature, together with low neutrophils, in association with a faster decline in insulin secretion and a younger age at onset [43]. Lastly, a recent article demonstrated that T1DE1 patients (<7 years old) exhibited a higher expression of the “do not eat me” signal CD47 as compared to the less aggressive T1DE2 endotype [44]. As CD47 expression is associated with inflammation, its upregulation could be the sign of a more pro-inflammatory environment in these pediatric patients. The idea of less aggressive forms of T1D being associated with age is further demonstrated by latent autoimmune diabetes in adults (LADA), a slow-onset endotype sharing aspects of both T1D and T2D, which can be considered an extreme T1E2 form. It is characterized by a later age at onset and an absence of insulin requirement at diagnosis, although sharing immunological features of childhood T1D, including the presence of aAbs [45]. We recently reported a description of the histopathology of a living LADA patient undergoing pancreatic surgery [46], which was characterized by significant β-cell loss and immune infiltrates comprising Foxp3^+^ regulatory T cells that are otherwise never observed in insulitis lesions. Together with the very late progression of this patient toward complete C-peptide loss and insulin dependency 8 years after surgery, this case report suggests that different degrees of immune regulation modulate the rate of β-cell decline. Collectively, all these findings confirm the existence of several forms of differentially aggressive T1D, linked to distinct autoimmune responses between age-related immunotypes and disease outcomes in new-onset T1D. To integrate diabetes heterogeneity and its multifactorial origins, McCarthy proposed a data-driven endotype definition [47] using a set of pathophysiological traits directly contributing to diabetes development and based on the strength of the signature of each component, taking into account genetic variations and history of environmental exposures. Applied to each patient, this code could help stratify endotypes sharing the same characteristics and predict diabetes evolution.

If age is a key parameter for discriminating endotypes, T1D endotypes can also be identified among aged-matched patients, as demonstrated by Arif et al. in new-onset T1D children [48]. The authors identified different immunophenotypes, with either multiple aAbs and an IFN-γ signature, or lower aAb numbers and a dominant IL-10 signature. Not only do these data suggest the existence of various endotypes among aged-matched and disease duration-matched patients, but the same associated signature was found in non-diabetic, multi-aAb^+^ siblings, further supporting the idea that such stratification could be used prior to diagnosis for targeted immunotherapies.

Another endotype stratification in line with personalized medicine in T1D relies on treatment responders versus non-responders. Regardless of whether the primary objectives were met or not, subgroups of patients with variable response rates are often recorded in clinical trials. For example, in the Protégé study that enrolled new-onset T1D patients receiving teplizumab or placebo, younger (8–11-year-old) patients had a greater C-peptide preservation than older age groups [49]. Similarly, patients defined by HLA-DR3/DR4 expression and ZnT8 aAbs seemed to be differentially responsive [15]. The discriminative impact of age was also observed in recently diagnosed patients upon abatacept therapy [50], where younger participants, exhibiting high circulating B-cell percentage with an activated signature, had the best clinical response; and upon rituximab treatment, where children had a greater therapeutic benefit [19]. This indicates that endotypes may condition the success or failure of immunotherapies.

The etiology underlying endotypes is most likely of both environmental and genetic origin, and identifying polymorphisms in genes involved in T1D could help further with stratifying these patients and optimizing their enrollment for therapy. For example, the *IL2RA* loci encoding for the IL-2 receptor α chain CD25 influences susceptibility to T1D [51,52], notably in patients diagnosed before 7 years of age [53], suggesting an association with a more aggressive endotype. This could be crucial in the context of IL-2-related therapies; dose-finding clinical trials using low-dose IL-2 administered to T1D adults [54,55] and recently diagnosed children [56] showed a dose-dependent increase in Treg numbers, raising hopes for the DIABIL-2 phase 2 trial currently ongoing (NCT02411253). Caution is advised, however, as low-dose IL-2 administered to T1D patients can concomitantly lead to the expansion of cytotoxic natural killer (NK) and CD8^+^ T cells [57]. In addition, the Treg expansion was heterogenous among individuals and over time. Thus, it is tempting to hypothesize that analyzing polymorphisms in the CD25 gene along other factors could help predict responders versus non-responders to such treatment, and enroll patients accordingly. Of note, similar results were found with the regulatory molecule CTLA-4, whose polymorphism is associated with human T1D risk, and a CTLA-4 splicing variant lacking CD80/86 ligand binding domain was associated in NOD mice with an increased susceptibility to the disease [58]. Single nucleotide polymorphisms (SNPs) in the gene encoding for the inhibitory receptor PD-1 (*PDCD1*) have also been identified and associated with increased T1D susceptibility [59,60], some of them being located at the binding site of RUNX1/AML1 transcription factors that are involved in cellular differentiation and regulation of hematopoiesis. Therapeutic exploitation of this pathway for controlling T1D could be achieved via tolerogenic dendritic cells (DCs), microbiota transfer, or grafting of engineered β cells expressing PD-L1; these options have been discussed elsewhere [61]. Deciphering patient-specific PD-1/PD-L1 expression and polymorphism could be of critical importance in T1D, since it was observed that T1D patients responding to anti-CD3 teplizumab therapy exhibit an exhausted T-cell phenotype characterized by PD-1 expression [62,63]. On the same line, expression of the indoleamine 2,3-dioxygenase 1 (IDO1) enzyme was shown defective in T1D children, with polymorphisms associated with T1D risk [64], and the treatment of peripheral blood mononuclear cells from diabetic patients exhibiting these risk haplotypes with the neutralizing anti-IL-6 mAb tocilizumab rescued IDO1 function. The use of NOD. *Ido1*^−/−^ mice demonstrated that the protective effect of tocilizumab on diabetes development was dependent on IDO1 [64], highlighting possible therapeutic implications. Finally, a clinical trial using subcutaneous GAD65 adjuvanted with aluminum hydroxide (GAD-alum) to treat new-onset T1D patients identified HLA-DR3/DQ2-positive individuals as better responders, with greater and dose-dependent response to treatment, as quantified by residual C-peptide secretion [65]. The crucial importance of HLA class II, and class I to a lesser extent, haplotypes in the susceptibility and severity of T1D has been extensively characterized [66,67], and these observations further suggest that HLA polymorphisms could impact the therapeutic benefit of antigen-specific approaches. Thus, screening patient HLA could not only help with defining their endotype as discussed above, but also with identifying responders to antigen-specific therapies, whose efficacy has so far been limited [68].

## 4. A Need for Biomarkers for Prognosis, Diagnosis, and Response to Therapy (“How”)

While stratifying patients based on their endotype and disease stage (“who and when”) could provide more personalized therapeutic options, this also highlights the crucial need for better biomarkers of T1D progression and response to immunotherapy, as current outcome measures for clinical trials are largely limited to metabolic (i.e., C-peptide preservation) and clinical criteria (i.e., insulin needs and glycemic control). Islet aAbs are useful to predict T1D risk, but not as surrogate biomarkers of efficacy in immune intervention trials, as their titers change slowly or not at all. T cells reactive to T1D-associated antigens are promising complementary biomarkers [69], as they could help stratifying patients based on their autoantigen reactivity profile to better gauge antigen-specific therapies. Using a combined peptidomics and transcriptomics approach, we have analyzed the peptidome of β cells exposed or not to inflammatory cytokines and identified known and novel epitopes, as well as neo-epitopes targeted by CD8^+^ T cells across HLA restrictions [70,71,72]. This kind of unbiased and comprehensive approach to antigen identification opens the possibility to measuring the overall autoimmune CD8^+^ T-cell “burden” of a given individual and provide better biomarkers for personalized intervention. In this perspective, we documented that islet-reactive CD8^+^ T cells circulate at the same frequency in the blood of T1D and healthy donors, and display a largely naïve phenotype, as defined by the co-expression of CD45RA and CCR7 [72,73,74]. This calls for the search of additional markers that could discriminate the autoimmune T cells of T1D versus healthy subjects (Figure 2). The dominant CD45RA^+^CCR7^+^ phenotype of islet-reactive CD8^+^ T cells may indeed correspond to less common memory-like T-cell subsets sharing surface naïve markers [75], such as the stem memory T cells (Tscm) that reportedly increased in T1D patients [76]. Similarly, recently described stem-like autoimmune progenitors expressing T-cell factor (TCF)1 and residing in pancreatic lymph nodes drive diabetes in NOD mice by giving rise to more differentiated effector cells that migrate to the pancreas and destroy β cells [77]. In line with this, β-cell reactive CD8^+^ T cells exhibit a unique DNA methylation signature with Tscm-like characteristics and both naïve and memory epigenetic programming [78]. A number of innate immunity-based signatures have also been identified in T1D patients and may provide additional biomarkers. For instance, neutrophils are involved in the early events of T1D pathogenesis [79], are decreased in the blood of T1D patients and aAb^+^ individuals, as compared to healthy donors [80], and exhibit impaired responses to cytokine priming [81]. In contrast, platelet–neutrophil aggregates, involved in neutrophil activation and migration, are increased in the blood of aAb^+^ and new-onset T1D children [82]. Finally, transcriptomics data demonstrated that not only these cells in at-risk individuals exhibit a type I IFN signature distinct from healthy donors, but also that such a signature is present in aAb-negative first-degree relatives of T1D patients [83]. In line with this, innate-derived inflammatory cytokines and chemokines were found upregulated in T1D patients as compared to healthy donors [84] and were able to discriminate between aAb-negative and aAb^+^ children [85]. Besides immune biomarkers, circulating biomarkers of β-cell stress or death are increasingly investigated. The detection of circulating DNA unmethylated at the *INS* locus as a biomarker of β-cell death did not hold its promise outside the acute setting of islet transplantation [86]. In contrast, the ratio of unprocessed proinsulin to mature insulin (measured as proinsulin/C-peptide ratio) is an extensively validated marker of β-cell stress, appearing before clinical onset in at-risk individuals [87] and persisting after diagnosis [88]. Other granule proteins undergoing a similar intermediate enzymatic processing such as islet amyloid polypeptide (IAPP) [89] and our recently described secretogranin-5, proconvertase-2, and urocortin-3 may provide complementary β-cell stress biomarkers [70,73,90]. Proteomics has extensively been used to identify other circulating biomarkers [91,92], such as insulin secretory granule and mitochondrial proteins, although their presence does not necessarily correlate with an increased β-cell death or with a more aggressive T1D evolution. Genetic risk scores have also been developed in order to predict T1D in at risk individuals, based on a combination of SNPs mostly from the HLA-DR/DQ region, but also from other HLA and non-HLA loci, and have been proven to be superior to HLA-based stratification alone [93,94]. Recently, a combined risk scoring system was proposed by taking into account the family history and the presence of aAbs together with the genetic risk score to predict T1D evolution in children, outperforming genetic risk score alone [95]. In line with this, the use of multi-omics analyses could further help identify the patient-integrated signature [96] not only to better define endotype characteristics and identify progressors and non-progressors, but also to stratify immunotherapy responders and non-responders during clinical trials [11].

Thus, integrating multiple immune, metabolic, and age parameters would offer more tailored immunotherapies. Using such biomarkers could moreover give clues on the outcome of the trial at early time points during immunotherapy, in addition to C-peptide preservation, which only gives a metabolic readout of treatment efficacy. Several indications support this idea. Edner et al. [97] demonstrated that the expression of the immunomodulatory markers ICOS and PD-1 on T follicular helper cells predicted the clinical response, as measured by C-peptide, in abatacept-treated T1D patients, which could help select patients that are the most likely to respond to this drug. In line with this, other studies reported that patients with a high inflammatory bias had a greater therapeutic response to abatacept [98]. Furthermore, whole-blood RNAseq analysis of abatacept-treated new-onset T1D patients demonstrated that B-cell related genes were augmented in non-responders, while neutrophil-related genes were upregulated in responders [50]. On the same line, in the follow-up study of the T1DAL trial [99], which treated recent-onset T1D patients with the immunomodulator alefacept, RNAseq was used to stratify patients as responders or non-responders, and allowed the identification within the responder group of a cluster of memory cells exhibiting an exhausted phenotype marked by TIGIT and KLRG1 expression. Similarly, a population of functionally unresponsive CD8^+^ T cells expressing TIGIT and KLRG1 was detected in T1D patients that responded best to teplizumab [100,101]. These data suggest that induction of T-cell exhaustion may be a common feature of positive responses to T-cell targeting immunotherapy in T1D and may thus be used as a biomarker predictive of increased responsiveness to treatment.

In summary (Table 1), biomarkers can be used not only to predict T1D progression, as is currently done with aAbs and HLA polymorphisms, but also to individually evaluate patient T-cell antigen specificity profile (autoimmune T-cell burden, stem memory-like signature) to assess β-cell stress (proinsulin/C-peptide ratio, role of secretory granule proteins) and to predict clinical response to immunotherapy (expression of immune checkpoint inhibitors, B-cell- and neutrophil-related signature). Screening these biomarkers could help stratify patients ahead of therapy and individually predict response to treatments.

## 5. Towards Personalized Combination Therapy (“What”)

The complexity of the immune networks driving T1D may explain the failure of single therapeutics to reverse or prevent disease course so far (with the exception of teplizumab delaying clinical progression). Hence, targeting multiple pathways may be required to induce a long-lasting control, with the final aim of offering a customizable multi-pronged therapy (Figure 3).

Several attempts have been made by combining immunomodulatory agents with complementary mode of action in preclinical models, but very few were translated to the clinic. Administration of rapamycin, an mTORc1-blocking drug inhibiting effector T cells but not Foxp3^+^ Tregs, with CD28 costimulatory molecule blockade [104] or with IL-2/anti-IL-2 immune complexes [105] demonstrated convincing capacities to delay diabetes in NOD mice. This latter drug combination was tested in T1D patients during a phase 1 trial, to favor Treg proliferation and function [106]. Despite the expected increase in Tregs, this treatment worsened C-peptide secretion, revealing that the tolerance-inducing effect of the combination therapy was unbalanced by the toxic effect of rapamycin on β-cell function/survival. Similarly, patients with new-onset T1D treated with ATG and G-CSF showed a relative preservation of the C-peptide secretion, together with increased Tregs. However, treatment with ATG alone was equivalent [107,108], lending credibility to a phase 2 trial currently ongoing (NCT04509791).

The success of these studies was modest overall and leaves room for improvement. Clinical trials proposing a combination of immunotherapies are currently ongoing in aAb^+^ at-risk patients, such as a phase 2 study using rituximab and abatacept (NCT03929601). However, combinatory agents and their window of application remain to be carefully selected. One could speculate that combining general targeting immunotherapy, such as anti-CD3 or anti-CD20 mAbs, with an antigen-specific approach to synergistically and selectively dampen autoreactive T-cell responses while improving antigen-specific tolerogenic mechanisms is the best way to start. In line with this idea, in 2010 the ITN–JDRF T1D Combination Therapies Assessment Group established a ranking of immune-based combinatory therapies for T1D [109], favoring combinations involving anti-CD3 mAb with antigen-specific (insulin, GAD) or anti-inflammatory (IL-1R/IL-1 blockade) therapies, which can be adjusted depending on the patient’s immune and antigen profile. In this line, a phase 1b/2a trial is currently evaluating the safety and tolerability of oral proinsulin-expressing *Lactococcus lactis* AG019 alone or in combination with teplizumab (NCT03751007). Despite solid evidence for gut dysbiosis in T1D, few trials have so far focused on microbiota-based therapies, mostly involving fecal transplantation [110], probiotics/prebiotics treatment [111], or administration of the short-chain fatty acids that mediate some beneficial effects of a healthy microbiota [112]. How these interventions mediate immunomodulatory or other effects remains to be further investigated. Tregs could also become an important component of combination therapies; although attractive, the clinical benefit obtained after infusion of autologous polyclonal Tregs to T1D patients was limited [113] but may be enhanced by immunomodulatory agents boosting Treg survival and function. To this end, a combination of autologous Tregs and low-dose IL-2 was tested in T1D patients [57]. Infused and endogenous Treg numbers were increased in treated patients, but this was accompanied by a substantial expansion and activation of cytotoxic CD8^+^ T cells and NK cells. Another, and probably safer, option, would be the use of β-cell antigen-specific Tregs based on the patient’s profile to strengthen the specificity and efficacy of such cell therapy. Another point to consider is the timing of administration of such treatment in the context of an ongoing autoimmune response, which may start by firstly blocking the immune responses with anti-CD3 or anti-CD20 mAb and associated inflammation (such as TNF, IL-1, type 1 IFN), and secondly by boosting regulatory mechanisms (Treg infusion, low dose IL-2, tolerogenic vaccination with islet antigens). A preclinical study in NOD mice [114] demonstrated that the infusion of BDC2.5 autoantigen-specific Tregs, but not polyclonal Tregs, after anti-CD3 therapy induced diabetes remission. The timing of intervention could be also crucial for B-cell targeting, as important changes have been reported in the B-cell compartment during T1D progression from aAb^−^ to established T1D [102], involving modifications of the frequency of transitional and anergic B cells, as well as their responsiveness to IL-21 and B-cell receptor (BCR) signaling that are unique for each stage. BCR signaling was also increased in responders to rituximab treatment. Combinatory options taking into account B-cell phenotype and functions at a certain time of T1D progression could help redirect them towards tolerance.

Another approach for personalized combination therapies consists of associating immunomodulating agents with drugs that help with regenerating the damaged β-cell number and/or function, which may then be suitable for patients with a more advanced disease. Liraglutide, an analog of glucagon-like peptide-1 that may improve β-cell mass and function, was combined with an anti-IL-21 neutralizing mAb in recent-onset T1D patients [115], leading to a moderate preservation of β cells. Clinical investigations are also ongoing with liraglutide in association with ex vivo expanded Tregs (NCT03011021). Importantly, the field of pancreatic β-cell replacement with engineered β cells, differentiated from induced pluripotent stem cells [116,117], is growing and could be combined with selected immunotherapies for the treatment of overtly diabetic (stage 3 and onwards) patients.

Finally, all clinical trials involving stage 3 T1D patients imply a combination therapy between the chosen agent and daily subcutaneous insulin administration that may potentially drive insulin antigen-specific effects to some extent. The immune correlates of this combination, i.e., whether this involves T-cell modifications in the insulin-reactive repertoire, remain to be determined.

Altogether, combination therapies require the consideration of the proper combination of drugs and sequence of treatments providing complementary/synergistic effects, on top of selecting the patients who are the most likely to respond to the treatment at the appropriate time of their disease progression.

## 6. Conclusions and Perspectives

Personalized therapeutics in T1D need to consider:(1)The “who”, i.e., disease endotypes, which may help identify subgroups of patients with a younger age at diagnosis and a more aggressive autoimmune disease, who would benefit best from intense and multi-hit immunomodulation, versus patients with a more indolent pathology and a slower evolution.(2)The “when”, i.e., the optimal stage of intervention, as patients before aAb appearance (stage 0) are the best suited for immunotherapies aimed at blocking the development of autoimmune responses, with tolerogenic introduction of targeted antigens before T-cell priming and epitope spreading occurs, and with fully preserved β-cell function and mass. For patients with clinically overt T1D (stage 3), β-cell replacement combined with tolerance maintaining protocols may provide a better option.(3)The “how” should take advantage of omics data to identify biomarkers that will assist with stratifying patients and identifying responders and the best combination therapy to use (the “what”).

These features will be key for designing more efficient clinical trials, and ultimately finding a treatment for T1D.

## Figures and Tables

**Figure 1 jpm-12-00542-f001:**
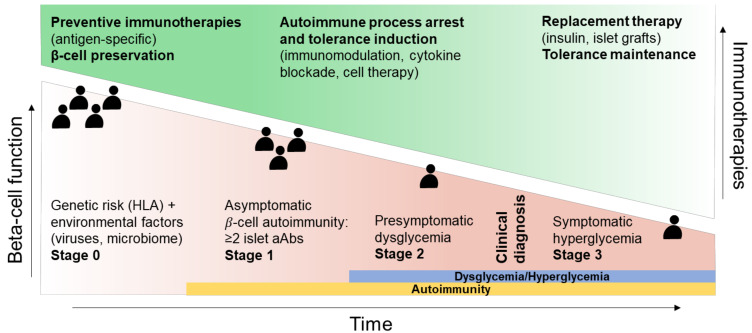
Stage-specific therapeutic intervention in T1D (type 1 diabetes). Stage 0 individuals, without aAbs (autoantibodies) but at risk for developing T1D owing to genetic and/or environmental triggers, are candidates for immunotherapies aiming at preventing T1D and preserving intact β-cell mass, such as antigen-specific and tolerance induction therapies. In subjects with stage 1 and stage 2 disease, characterized by the presence of one or more aAbs and a progressive dysglycemia, therapies aiming at dampening β-cell destruction and the engaged autoimmune process while favoring tolerance-promoting mechanisms (T-cell, B-cell depletion/inhibition/exhaustion, blockade of inflammatory cytokine signaling, adoptive Treg cell therapy) should be preferred. Therapeutic intervention focusing on replacing endogenous β cells by grafting pancreatic islets or in vitro differentiated β cells, together with tolerance-promoting drugs, should be reserved to stage 3 patients with clinically overt T1D and an advanced decline in β-cell number and function. Abbreviation: HLA, human leukocyte antigen.

**Figure 2 jpm-12-00542-f002:**
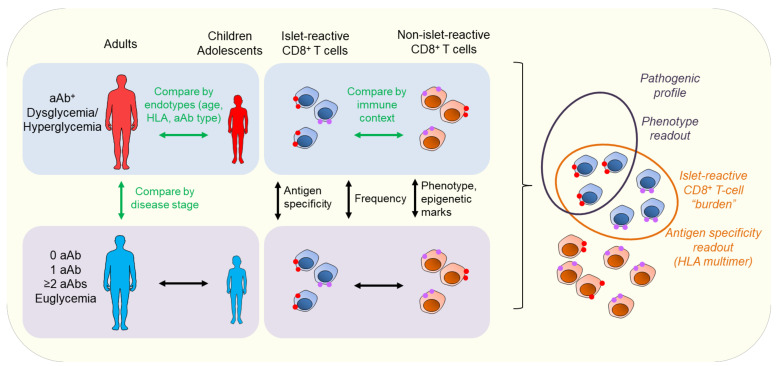
Biomarker identification for personalized therapy in T1D (type 1 diabetes). Stratification of T1D patients can be achieved by comparing disease stage, from at-risk individuals to patients with clinical disease, based on the presence of aAbs (autoantibodies), their type, and number, together with a progressive dysglycemia and hyperglycemia. This is completed by further stratifying the patient according to their endotypes, based on criteria including age at onset, first aAb at seroconversion, and HLA (human leukocyte antigen) typing. In parallel, comparing the overall immune context, such as innate and adaptive signature, gives cues on the disease aggressiveness and the ongoing autoimmune process. In particular, islet-reactive CD8^+^ T-cell frequency, phenotype, epigenetic signature, and overall burden may contribute to categorizing patient autoimmunity.

**Figure 3 jpm-12-00542-f003:**
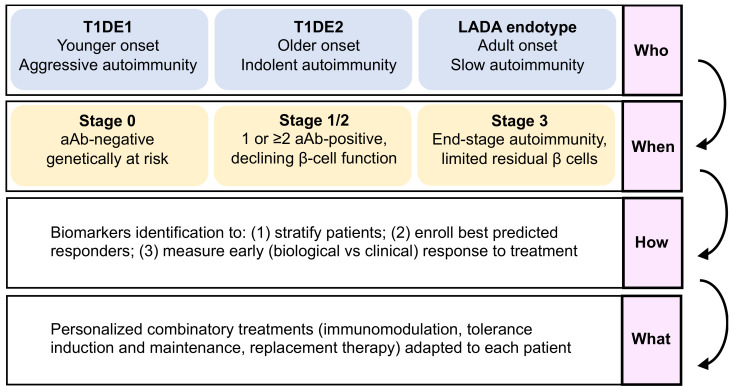
The who, when, how, and what paradigm for T1D (type 1 diabetes) personalized therapy. The first layer of selection for individualized intervention consists in stratifying T1D endotypes, in order to adapt therapies to distinct immunopathology profiles, such as T1DE1, characterized by a younger age at onset and an aggressive phenotype, and patients with a T1DE2, presenting indolent immune responses and older age at onset, versus slow progressive and initially non-insulin-requiring latent autoimmune diabetes in adults (LADA) endotype. The second layer focuses on selecting an appropriate window of intervention along the natural history of the disease, given the immunological discrepancies between stages. Such stratification nonetheless relies on biomarker screening to identify patient subtype and stage of disease, but also to predict best responders to a therapy as well as measure early responses to therapies. Altogether, such a strategy may be applied to offer a personalized treatment or combination of treatments adapted to each patient’s pathobiological characteristics of T1D progression.

**Table 1 jpm-12-00542-t001:** Biomarkers identifying responders and non-responders to immunotherapies in T1D patients.

Biomarker	Drug	Patients Stage	Associated Phenotype	Reference
ICOS and PD-1 expression on T follicular helper	abatacept	New-onset T1D	Predict decreased response to treatment	[97]
B-cell-related genes	abatacept	New-onset T1D	Increased in non-responders	[50]
Neutrophil-related genes	abatacept	New-onset T1D	Increased in responders	[50]
TIGIT and KLRG1 on CD8^+^ T cells	alefacept	New-onset T1D	Increased in responders	[99]
TIGIT and KLRG1 on CD8^+^ T cells	teplizumab	New-onset T1D	Increased in responders	[100,101]
Low innate inflammatory signature	abatacept	New-onset T1D	Greater responders	[98]
HLADR4^+^ HLA-DR3^−^	teplizumab	At risk, ≥2 aAbs	Increased in responders	[15]
ZnT8 autoantibodies	teplizumab	At risk, ≥2 aAbs	Decreased in responders	[15]
PD-1 expression on CD4^+^ and CD8^+^ T cells	teplizumab	New-onset T1D (8 weeks)	Increased in responders	[62,63]
BCR signaling	rituximab	Established T1D	Increased in responders	[102]
Heterogenous T-cell signature	rituximab	New-onset T1D	Decreased efficacy	[103]

Abbreviations: T1D, type 1 diabetes, HLA, human leukocyte antigen; aAbs, autoantibodies; BCR, B-cell receptor.

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
