# Peer review of "Personalized Immunotherapies for Type 1 Diabetes: Who, What, When, and How?"

_jpm, 2022, doi:10.3390/jpm12040542_

Round 1
Reviewer 1 Report
The review summarized the finished and ungoing trials of immunotherapies in T1D and discussed how to make a personalized strategy to increase therapeutic efficacy from four dimensions, which is important in field of precised treatment for T1D. There are several minor concerns:
- Innate immunity is also involved in the pathogenesis of T1D and is regarded as the initial events of immune disorders in islets. Inflammatory signatures, cytokines and chemokines in innate immune cells such as neutrophils, macrophages, which could be used as biomarkers in prediction and early diagnosis of T1D should be discussed more.
- Besides HLA genotypes, genetic risk score (GRS) based on HLA and non-HLA genes predict the risk of T1D and C-peptide deterioration. Whether GRS is a method to identify high risk individuals for intervention can be taken into consideration.
- Some studies include LADA in the definition of adult-onset T1D, however, the differences between adult-onset T1D in endotype TIDE2 and LADA should be discussed.
- What is the research status of gut microbiome-related Immunotherapies? Only one trial NCT03751007 is mentioned.
Author Response
The review summarized the finished and ongoing trials of immunotherapies in T1D and discussed how to make a personalized strategy to increase therapeutic efficacy from four dimensions, which is important in field of precised treatment for T1D.
We thank the Reviewer for her/his general comment on the interest of our manuscript in the field of personalized treatments for T1D.
There are several minor concerns:
- Innate immunity is also involved in the pathogenesis of T1D and is regarded as the initial events of immune disorders in islets. Inflammatory signatures, cytokines and chemokines in innate immune cells such as neutrophils, macrophages, which could be used as biomarkers in prediction and early diagnosis of T1D should be discussed more.
The point is well taken. We have now added a paragraph in the section “A need for biomarkers for prognosis, diagnosis and response to therapy (“how”)”, lines 228-235, describing the use of innate immunity related biomarkers that could be used to predict T1D. In the original MS version, we also discussed the importance of type I IFN signatures indicative of innate responses.
- Besides HLA genotypes, genetic risk score (GRS) based on HLA and non-HLA genes predict the risk of T1D and C-peptide deterioration. Whether GRS is a method to identify high risk individuals for intervention can be taken into consideration.
We have added a paragraph in the same section, lines 243-248, that describes the potential use of genetic risk score for the prediction of T1D among other immune and beta-cell based biomarkers.
- Some studies include LADA in the definition of adult-onset T1D, however, the differences between adult-onset T1D in endotype TIDE2 and LADA should be discussed.
Indeed the wider clinical heterogeneity of adult T1D is another point to consider. T1DE2 does not resume to LADA, and LADA may be considered an extreme case of T1DE2 with a particularly mild autoimmunity. A section has now been added in the Endotypes “Who ?” section, lines 155-159, also quoting our recent report which describes the pancreas histopathological features of a living LADA donor (reference #46).
- What is the research status of gut microbiome-related Immunotherapies? Only one trial NCT03751007 is mentioned.
The clinical trial mentioned in the review (NCT03751007) is rather about combination of immunotherapies than microbiota. Although gut dysbiosis has been largely described in T1D, its therapeutic applications have so far been limited to non-immune interventions including the use of probiotics and fecal transfer treatments, which are outside the scope of this manuscript. We have however now mentioned to microbiota-related therapies in the section “Towards personalized combination therapy (“what”)”, lines 300-303.
Reviewer 2 Report
The review by Deligne et al. is a comprehensive summary of needs and opportunities of personalized immunotherapies for Type 1 Diabetes. The topic of the review is of high importance, since the disease heterogeneity of T1D, including multiple immune, metabolic and clinical parameters, is still a considerable hurdle for successful immunotherapies to prevent or revert Type 1 Diabetes. The structure of the review (who, what., when and how) is well chosen and quite helpful to understand the complexity of what has to be addressed for successful personalized immunotherapies in Type 1 Diabetes. The authors discuss the most important studies with regard to disease stage, disease endotypes, biomarkers and personalized combination therapies and point out the knowledge gaps and how they could potentially overcome.
The references are appropriately chosen, quite recent and only a few additional references should be added, as indicated below:
- With regard to the decreased life expectancy (line 43) the following citation should be included:
Rawshani, A., et al., Excess mortality and cardiovascular disease in young adults with type 1 diabetes in relation to age at onset: a nationwide, register-based cohort study. Lancet, 2018. 392(10146): p. 477-486. - With regard to the Teplizumab studies the following citation should be included:
Sims, E.K., et al., Teplizumab improves and stabilizes beta cell function in antibody-positive high-risk individuals. Sci Transl Med, 2021. 13(583) - With regard to the oral insulin studies the following citation should be included:
doi: 10.1007/s00125-020-05376-1
The figures are a relevant and helpful addition to the review. However, they would benefit from an additional revision especially regarding a reduction of the text and partially the better use of colors and icons to increase clarity and comprehensibility.
Overall, based on the abovementioned points, this review is an important contribution to the field and will be of interest for the readership of the Journal of Personalized Medicine.
Author Response
The review by Deligne et al. is a comprehensive summary of needs and opportunities of personalized immunotherapies for Type 1 Diabetes. The topic of the review is of high importance, since the disease heterogeneity of T1D, including multiple immune, metabolic and clinical parameters, is still a considerable hurdle for successful immunotherapies to prevent or revert Type 1 Diabetes. The structure of the review (who, what., when and how) is well chosen and quite helpful to understand the complexity of what has to be addressed for successful personalized immunotherapies in Type 1 Diabetes. The authors discuss the most important studies with regard to disease stage, disease endotypes, biomarkers and personalized combination therapies and point out the knowledge gaps and how they could potentially overcome.
We thank the Reviewer for her/his positive evaluation.
The references are appropriately chosen, quite recent and only a few additional references should be added, as indicated below:
- With regard to the decreased life expectancy (line 43) the following citation should be included:
Rawshani, A., et al., Excess mortality and cardiovascular disease in young adults with type 1 diabetes in relation to age at onset: a nationwide, register-based cohort study. Lancet, 2018. 392(10146): p. 477-486. - With regard to the Teplizumab studies the following citation should be included:
Sims, E.K., et al., Teplizumab improves and stabilizes beta cell function in antibody-positive high-risk individuals. Sci Transl Med, 2021. 13(583) - With regard to the oral insulin studies the following citation should be included:
doi: 10.1007/s00125-020-05376-1
All the above-mentioned articles have been added and quoted accordingly:
- Rawshani, A., et al.: reference #9, Introduction, line 44
- Sims, E.K., et al.: reference #103, A need for biomarkers for prognosis, diagnosis and response to therapy (“how”) section, line 266 and Table 1.
- Assfalg R. et al. : reference #27, Disease stage (“when”), lines 92-95
The figures are a relevant and helpful addition to the review. However, they would benefit from an additional revision especially regarding a reduction of the text and partially the better use of colors and icons to increase clarity and comprehensibility.
To improve the figure clarity, we have removed some text from Figures 1 and 3 and modified the colors.
Overall, based on the abovementioned points, this review is an important contribution to the field and will be of interest for the readership of the Journal of Personalized Medicine.
Thank you again for these constructive comments to help us improving the manuscript.
Round 2
Reviewer 1 Report
I have no further comments.